

**Enhancing flood forecasting reliability in data-scarce regions with a distributed hydrology-guided neural network framework**

Confidence Duku[1,2]
[1]Wageningen Environmental Research, Team Climate Resilience, Wageningen University & Research,
Wageningen, The Netherlands
[2]Earth Systems and Global Change Group, Wageningen University & Research, Wageningen, The
Netherlands
*Correspondence to:* Confidence Duku(confidence.duku@wur.nl)
**Abstract**
Flood early warning systems are critical for reducing disaster impacts, yet their effectiveness remains
limited in data-scarce regions such as Africa and South America. Existing global platforms—including
GloFAS and the Google Flood Hub—exhibit low reliability in these areas, particularly for rare flood events
and under strict timing constraints. Here, I demonstrate the potential of a distributed, hydrology-guided
neural network framework, Bakaano-Hydro, to enhance flood forecasting reliability in data-scarce regions.
The proposed framework integrates process-based runoff generation, topographic routing, and a Temporal
Convolutional Network for streamflow simulation. Using a hindcast-based evaluation across 470 gauging
stations from 1982 to 2016, I benchmark Bakaano-Hydro's flood detection skill against GloFAS and Google
AI model across multiple return periods (1-, 2-, 5-, and 10-year) and timing tolerances (0–2 days). Results
show that Bakaano-Hydro consistently achieves higher Critical Success Index (CSI), lower False Alarm Rate
(FAR), and higher Probability of Detection (POD), even under exact-day (0-day) timing constraints. Its
median CSI scores at 0-day tolerance exceed or match those of GloFAS and Google AI model under more
lenient timing thresholds. These performance gains are statistically significant across diverse hydroclimatic
regions, including arid and tropical basins, demonstrating the model's spatial generalization capacity. By
coupling physical realism with machine learning generalizability, Bakaano-Hydro provides a reliable,
interpretable, and open-source tool for enhancing flood forecasting in regions most vulnerable to climate
extremes and least equipped with observational infrastructure.
**1. Introduction**
Floods are the most frequent and widespread natural disaster globally, accounting for more than 40% of
all weather-related hazards and affecting over 1.5 billion people between 2000 and 2020 alone (Yaghmaei,
2020). In 2021, floods caused an estimated $105 billion USD in economic losses worldwide, with Africa,



South-East Asia and Oceania bearing a disproportionate share of fatalities and damage due to weaker
infrastructure and limited adaptive capacity (AON, 2023). The burden is particularly severe in Africa, where
more than 75 million people live in high flood-risk zones and simultaneously face extreme poverty
(Rentschler et al., 2022). Recent catastrophic events in Nigeria, Sudan, South Africa, and Mozambique
underscore how recurrent flooding exacerbates food insecurity, damages livelihoods, and displaces
communities—often with cascading socio-economic consequences.
As climate change intensifies extreme precipitation events, and urbanization continues to encroach upon
floodplains (Mazzoleni et al., 2022; Tellman et al., 2021), the need for accurate and timely early warning
systems becomes increasingly urgent. For example, the last twenty years have seen the number of major
floods more than double (Yaghmaei, 2020). Flood early warning systems can reduce mortality by up to 43%
(WMO, 2013) and cut economic losses by as much as 50% (Pilon, 2002; Rogers & Tsirkunov, 2011), yet
these benefits are unevenly distributed. Enhancing the performance of early warning systems in
developing countries to levels comparable with high-income settings could prevent an estimated 23,000
deaths annually (Hallegatte, 2012). However, many of the most flood-prone and affected regions lack
robust national flood early warning systems. This limits governments' ability to anticipate and respond to
disasters in a timely manner. As a result, disaster response agencies—including national emergency
authorities, humanitarian organizations, and international development partners—often depend on global
flood forecasting platforms such as the Global Flood Awareness System (GloFAS) (Alfieri et al., 2013; Alfieri
et al., 2020; Harrigan et al., 2023), part of the Copernicus Emergency Management Service and the Google
Flood Hub (Nearing et al., 2024; Nevo et al., 2022). These systems offer publicly accessible, transboundary
early warnings and have become essential tools for operational response in data-scarce regions. GloFAS,
couples weather forecasts with the LISFLOOD model (De Roo et al., 2000), a physically based, fully
distributed hydrological model in generating flood forecasts. By contrast, the Google AI model incorporates
weather forecasts into a global data-driven hydrological model (Kratzert et al., 2019), which simulates river
flows and potential flood events.
Despite recent advances, the skills of these models in data-scarce regions, are limited and consequently
forecasts remain unreliable. An evaluation of GloFAS and the Google AI model performance across flood
events with varying return periods (1-, 2-, 5-, and 10-year) from 1984 to 2022 revealed notably lower
median performance in Africa and South America compared with Europe and North America (Nearing et
al., 2024). The skill gap is especially pronounced for rarer flood events, where the models' reliability
deteriorates further. The persistent low reliability is particularly concerning given the heightened



vulnerability of communities in these regions, where limited infrastructure, higher poverty rates, and lower
adaptive capacity amplify the social and economic consequences of flooding (Fox et al., 2024; Sauer et al.,
2024). A key factor underlying poor model performance is the sparse, fragmented observational networks
in these regions, compounded by issues of data quality, restricted accessibility, and infrastructural
limitations (Grimes et al., 2022). The process-based modelling approach employed by GloFAS typically
requires extensive calibration to individual basins and struggles to generalize to ungauged areas. While
Google AI model's data-driven approach can potentially learn representations across basins, its lumped
modeling framework—relying on area-weighted averages of climatic forcings, vegetation, and soil
attributes—cannot fully capture localized hydrological processes.
Several methodological advances have been proposed to address the limitations of physically based
hydrological models in data-scarce regions. Regionalization techniques, which transfer calibrated
parameters from data-rich to poorly monitored catchments, have been widely adopted, but their
performance remains limited in climatically heterogeneous or poorly instrumented areas (Beck et al., 2016;
Guo et al., 2021; Pagliero et al., 2019). Bias correction of meteorological forcings and streamflow, has also
been proposed to improve the accuracy of forecasts (Crochemore et al., 2016; Tanguy et al., 2025).
However, bias correction requires long, high-quality historical observations to calibrate correction
functions, limiting their applicability in regions with sparse or unreliable in-situ data. Similarly, a promising
direction is data assimilation, particularly the integration of satellite-derived hydrological variables—such
as soil moisture, snow cover, and river levels—into forecasting models to correct state variables and
improve flood prediction in regions with limited in-situ observations. (Alfieri et al., 2022; Emerton et al.,
2016; Wongchuig et al., 2024). However, in data-limited regions, the effectiveness of data assimilation is
often constrained by the coarse resolution and uncertainty of satellite products, scale mismatches, and the
need for careful tuning and error characterization in the absence of reliable ground-based data. In parallel,
data-driven models, particularly deep learning architectures such as Long Short-Term Memory (LSTM)
networks, have demonstrated strong performance in capturing complex, nonlinear rainfall–runoff
relationships without requiring explicit process parameterization (Feng et al., 2020; Hunt et al., 2022;
Kratzert et al., 2018). These models have contributed to advances in streamflow prediction and, by
extension, flood forecasting skill. However, they are often developed using lumped or basin-aggregated
inputs, lack spatially explicit representation of hydrological processes, and performance tends to degrade
with increasing basin size (Hunt et al., 2022). Notably, incorporating physical constraints into data-driven
models has been shown to improve predictive accuracy and enhance robustness across diverse
hydrological settings (Kratzert et al., 2019). Additionally, accounting for spatial variability in inputs,





particularly rainfall, can further improve the performance of lumped data-driven models (Wang & Karimi,
2022), highlighting the value of physically informed machine learning approaches that combine data-driven
flexibility with process-based realism.
To address the limitations of current flood forecasting systems in data-scarce regions, here, I demonstrate
the potential of a distributed, hydrology-guided neural network modelling framework to significantly
enhance forecasting reliability across Africa and South America. The modelling framework uniquely
integrates physically based hydrological principles with the generalization capacity of machine learning in
a spatially explicit and physically meaningful way. The framework referred to as Bakaano-Hydro is
benchmarked against GloFAS and Google AI model. The name Bakaano comes from Fante, a language
spoken along the southern coast of Ghana. Loosely translated as "by the river side" or "stream-side", it
reflects the lived reality of many vulnerable riverine communities across the Global South - those most
exposed to flood risk and often least equipped to adapt. In this study, retrospective simulations (hindcasts)
are used to evaluate and compare Bakaano-Hydro performance against GloFAS and Google AI model. While
not true forecasts, these hindcasts serve as a standard proxy for assessing forecasting reliability, as
commonly done in forecast evaluation and earlier research (e.g. Alfieri et al., 2013; Nearing et al., 2024).
Accordingly, I use the term 'forecasting reliability' to refer to the skill of models in reproducing observed
flood events across return periods and timing tolerances using historical input data. The Bakaano-Hydro
workflow consists of three stages: (1) the estimation of spatially distributed runoff using a land surface
model designed to capture spatiotemporal variability; (2) routing of this runoff using a topographic flow
direction algorithm; and (3) prediction of daily streamflow using a temporal neural network trained on
routed runoff. Observed streamflow data from 643 gauging stations across Africa and South America are
used for training and evaluation. A single model is employed and trained jointly across Africa and South
America to enhance spatial generalization. Flood return periods (1-, 2-, 5-, and 10-year events) are
computed from observed streamflow at each station and consistently applied across Bakaano-Hydro,
GloFAS, and Google AI model outputs. Critical Success Index (CSI), False Alarm Ratio (FAR), and Probability
of Detection (POD) metrics are calculated under multiple flood timing tolerances (0, 1, and 2 days) to
evaluate both detection accuracy and temporal precision.





## 2. Methods

## 2.1 The Bakaano-Hydro modelling framework and data

The selection of river basins in this study (Figure.1) was guided by the need to evaluate flood forecasting skill across a hydroclimatically and geographically diverse set of regions, representative of key challenges in data-scarce environments. The basins span a wide range of climatic zones—from humid tropics (e.g., Amazon, Congo, Niger) to semi-arid and arid systems (e.g., Volta, Orange, Brazil Northeast, and Rio Colorado)—and vary substantially in terms of flood regime, seasonality, land use, and topographic complexity. Several basins (e.g., Amazon, Congo, and Niger) are among the largest in the world and are subject to recurrent fluvial flooding, while others (e.g., West Africa coastal basins, Brazilian coastal and interior basins) represent smaller, flashier catchments prone to localized flood events. Importantly, all selected basins are located in regions with limited in-situ hydrometeorological infrastructure, where global forecasting systems often perform poorly due to sparse calibration data and limited representation of local hydrological processes. This selection enables a robust, spatially explicit evaluation of model generalization across diverse flood-generating mechanisms, data conditions, and socio-environmental contexts in the Global South.

In this implementation of the Bakaano-Hydro framework, the VegET method was first employed to estimate total runoff. VegET, primarily designed for actual evapotranspiration estimation, also incorporates key hydrological processes such as rainfall interception, runoff generation, and soil moisture storage (Senay, 2008; Senay et al., 2023). The method operates on a daily time-step, where actual evapotranspiration is estimated as a function of the Normalized Difference Vegetation Index (NDVI) and reference evapotranspiration. Rainfall interception is estimated as a function of tree cover, herbaceous cover, and bare soil fractions, while runoff is computed using a saturation-excess approach, whereby excess soil water beyond the field capacity is considered unavailable for plant uptake in the root zone. Daily meteorological inputs, including precipitation and temperature (minimum, maximum, and mean), were obtained from the CHELSA-W5E5 database a bias-adjusted high resolution global climate dataset derived from CHELSA (Climatologies at High Resolution for the Earth's Land Surface Areas) and adjusted using W5E5 reference data (Karger et al., 2023; Karger, 2021). Data availability was limited to the period 1981–2016. NDVI and fractional tree and herbaceous cover data spanning 2001–2016 were obtained from Moderate Resolution Imaging Spectroradiometer (MODIS) remote sensing products (Didan, 2021; DiMiceli et al., 2015). MODIS-NDVI products are unavailable for periods before 2001. Following the VegET procedure, a daily mean





climatology of NDVI  was established with linear interpolation. Tree cover and herbaceous cover fractions
were also obtained from MODIS (DiMiceli et al., 2015) for the period 2001 – 2016. MODIS tree cover and
herbaceous cover products are unavailable for periods before 2001. Annual mean tree cover and
herbaceous cover fractions were then computed over this time period and were used in runoff estimation.
Soil properties, including wilting point, field capacity, and saturation, were obtained from SoilGrids
database, which is a global soil information system that provides predictions of standard soil properties at
multiple depths and at high resolution (Poggio et al., 2021). Digital Elevation Model (DEM) was sourced
from HydroSHEDS database (Lehner et al., 2008), which provides hydrologically corrected topographic
data. All input datasets were resampled to 1km$^2$ resolution, matching the spatial resolution of climate data.
Total runoff was computed for the 1981–2016 period, consistent with the availability of climate input data.
Potential evapotranspiration as part of VegET was estimated using the Hargreaves equation (Hargreaves &
Samani, 1985). Further details on the VegET model and its parameterizations are available in Senay et al.

13   (2023).

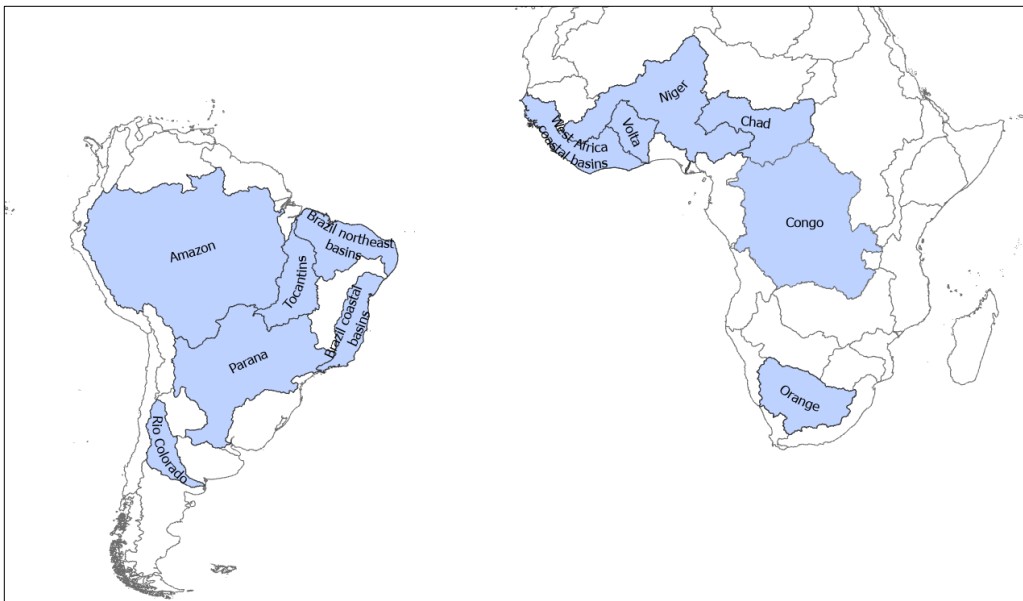

**Figure 1.** Basins for which observed streamflow data were used in training, evaluating and computing
reliability metrics of Bakaano-Hydro. Basin delineation were obtained from HydroSHEDS (Lehner et al.,
2008). Brazil northeast basins,  Brazil coastal basins and West Africa coastal basins are not recognized
names but refer to group of smaller basins in the named locations that have been aggregated for the
purpose of this study.



The second phase of Bakaano-Hydro is the flow routing phase which bridges empirical or physically based
surface runoff generation with data-driven streamflow prediction. In this analysis, daily runoff was routed
through the river channel network using the weighted flow accumulation method based on the multiple
flow direction approach (Quinn et al., 1991). This routing scheme distributes flow from each cell to up to
eight neighboring cells, with partitioning proportional to the elevation gradient between cells. For each of
the 470 gauging stations, routed runoff time-series were extracted from the river network to facilitate
further analysis. Routed runoff time-series are extracted based on the longitude and latitude of the gauging
station. To allow for distortions in geometry and provide tolerance, the coordinates are snapped to the
river network.
The final phase of Bakaano-Hydro involves the application of a deep learning model to simulate streamflow.
To capture the complex, nonlinear relationships between routed runoff and streamflow a Temporal
Convolutional Network (TCN) was employed. TCNs are a class of deep learning models designed for
sequence modeling tasks and are based on one-dimensional causal convolutions with dilated filters,
allowing the network to efficiently capture long-range temporal dependencies while maintaining a fixed
input length and avoiding information leakage from future timesteps (Bai et al., 2018). Compared to LSTM
networks, which rely on recurrent structures and hidden states passed sequentially through time, TCNs
use convolutional layers that process entire sequences in parallel. This leads to faster training times, greater
stability during optimization, and better scalability to long input sequences, especially in large
spatiotemporal datasets. Additionally, TCNs avoid issues commonly associated with recurrent networks,
such as vanishing gradients and limited interpretability of memory cells. In the context of Bakaano-Hydro,
TCNs are particularly advantageous because they can efficiently model multi-scale temporal patterns in the
routed runoff time series across hundreds of locations, while enabling the use of a shared model
architecture for diverse river basins. Their ability to learn temporal features over fixed receptive fields,
combined with their parallelizability and robustness, makes TCNs a suitable and scalable choice for
simulating streamflow from spatially distributed runoff inputs in data-limited, flood-prone regions. In
Bakaano-Hydro, the TCN architecture is configured to utilize a hindcast sequence of 365 days to predict
daily streamflow. The architecture consists of two input branches, both employing TCN layers to process
dynamic and static variables. The first input branch processes nine variables—three dynamic and six
static—standardized across all 643 stations in Africa and South America using z-score normalization.
Dynamic variables include routed runoff extracted at gauging stations, augmented by dividing by the
upstream contributing area (in grid cells) and the depth-to-water index. Static variables include the
upstream contributing area, depth-to-water index, and sine-cosine transformations of station latitude and



longitude to encode spatial periodicity. The static variables were repeated across the entire duration of the
dynamic variables. The second branch processes station-specific routed runoff data, scaled by the
maximum daily routed runoff across all stations within the basin and log-transformed to reduce skewness.
TCN layers employ dilation rates of 2, 4, 8, 16, 32, 64, 128, and 256, capturing temporal dependencies
across multiple timescales. The outputs from the two TCN branches are concatenated and further
processed through dense layers to establish full connectivity, integrating global and station-specific runoff-
streamflow relationships. Observed streamflow data were sourced from the Global Runoff Data Centre
(GRDC) (GRDC, 2025) and included gauging stations across Africa and South America with at least five years
of data. The training period covered 1989–2016, while model validation was conducted for the 1982–1988
period. Only hydrological stations with observed streamflow record of at least five years were used in the
training of the model.

## 2.2 Benchmark data from GloFAS and Google AI model

The Google AI model builds on previous LSTM-based nowcasting approaches by employing an encoder–
decoder LSTM architecture and area-weighted averages over basin polygons over the total upstream area
of each gauge or prediction point. The model was trained using data from 5680 stations across the globe
(Nearing et al., 2024). The Google AI model data used here were derived from a full model run
encompassing all stations, rather than the cross-validation splits reported in Nearing et al. (2024). The
streamflow predictions are right-labeled, meaning the predicted value at day $t$ corresponds to the
observation at day $t-1$. To ensure consistency, we relabeled their predictions to match the observation
timestamps used in our dataset.
GloFAS data are from GloFAS version 4, which is the latest version as at the time of submission. As part of
GloFAS, the LISFLOOD OS model, implemented at a 0.05-degree quasi-global resolution  was calibrated
using in-situ discharge data from 1,996 stations with drainage areas of at least 500 km² and observation
records post-1980. These stations were calibrated using the Distributed Evolutionary Algorithm for Python
(Fortin et al., 2012). Parameter values for ungauged catchments estimated through regionalization
(Grimaldi, 2024).

## 2.3 Estimation of flow thresholds for multiple return periods

A common subset of 470 hydrological stations was identified across Bakaano-Hydro, GloFAS, and Google
AI model to enable consistent model evaluation. Return periods for these stations were estimated using
observed streamflow records from the GRDC database. The estimation followed a modular approach based



on the U.S. Geological Survey (USGS) Bulletin 17C guidelines (England Jr et al., 2018), incorporating
procedures for annual peak extraction, low outlier treatment, and frequency distribution fitting. Flow
thresholds corresponding to the 1-, 2-, 5-, and 10-year return periods were computed and applied
uniformly across model outputs to evaluate flood detection skill.. The methodology comprises three main
components: (1) extraction of peak flows, (2) identification and treatment of potentially influential low
floods (PILFs), and (3) return period estimation using either parametric or empirical approaches. Peak flows
were extracted from continuous daily discharge records using the annual maximum series (AMS) method.
This method identifies the single highest discharge in each water year and is the standard recommended
by England Jr et al. (2018). The water year is defined such that it ends in September, aligning with typical
hydrological year definitions in the Northern Hemisphere. A minimum data completeness threshold of 50%
of the expected daily observations per year was imposed to ensure robustness. To mitigate the influence
of potentially impactful low floods (PILFs), we employed the Grubbs-Beck Test (GBT) as described in
England Jr et al. (2018). This test detects low outliers by comparing observations against a lower threshold
derived from sample mean and standard deviation, using critical values from a tabulated KN statistic. Data
points below the threshold were flagged as PILFs and subsequently censored using a left-censoring
approach during distribution fitting. The PILF threshold was defined as the midpoint between the largest
outlier and the smallest non-outlier. For stations with insufficient sample size to run the GBT (i.e., fewer
than 10 observations), no censoring was applied.
The primary method employed was the Generalized Expected Moments Algorithm (GEMA) for fitting a log-
Pearson Type III distribution, following the full procedure described in England Jr et al. (2018). This iterative
method jointly estimates distribution parameters (shape $\alpha$, scale $\beta$, and location $\tau$) from censored and
uncensored data by aligning sample and distribution moments. Convergence was achieved when the L1
norm of moment differences fell below a threshold of (1e−10). PILFs were incorporated as left-censored
values below the GBT-derived threshold. This method is particularly suited to datasets with both low
outliers and log-normality. In cases where the user explicitly disabled GEMA or when GEMA failed to
converge, we applied standard moment-based fitting of a log-Pearson III distribution without PILF
treatment. This approach estimates distribution parameters directly from the log-transformed sample
mean, standard deviation, and skewness (England Jr et al., 2018). As a fallback for short time series or when
moment-based fitting was numerically unstable, we estimated return periods using a log-log linear
regression between observed flow magnitudes and empirical exceedance probabilities. Exceedance
probabilities were computed using the Weibull plotting position formula. While this empirical method lacks



the theoretical rigor of the log-Pearson III distribution, it provides a practical option for very limited
samples.

### 2.4 Estimation of flood forecasting reliability metrics

CSI, FAR and POD were used as a measure of reliability and were computed over the period 1982 – 2016.
The streamflow thresholds were applied to the simulated streamflow data from GloFAS, Google AI model
and Bakaano-Hydro. Based on this, a model's prediction of an event with a given return period was
considered correct if both the modeled and observed hydrographs exceeded their respective return period
threshold flow values within defined timing tolerance. We used three flood timing tolerance which define
the permissible temporal deviation between observed and predicted flood events when evaluating model
performance. In the 0-day timing tolerance, a predicted flood event must occur on the exact day of the
observed event to be considered a match; the 1-day timing tolerance allows for a match if a predicted flood
occurs within one day (before or after) of the observed event and the 2-day timing tolerance extends the
matching period to two days. CSI is computed as $TP/(TP + FP + FN)$; POD is $TP/(TP + FN)$ and FAR is
computed as $FP/(TP + FP)$ where TP is True Positive describing correctly predicted flood events for a
specified return period and flood timing tolerance; FP is False Positives describing predicted flood events
that are not in observed data; and FN is False Negatives describing flood events in the observed data but
were not predicted.

### 3. Results

### 3.1 Performance of Bakaano-Hydro

Out-of-sample validation (Figure 2) demonstrates that the Bakaano-Hydro framework achieves strong and
spatially consistent performance across major basins in Africa and South America. Median values of Nash-
Sutcliffe Efficiency (NSE), Kling-Gupta Efficiency (KGE) exceed 0.5 in most basins, indicating robust
agreement between simulated and observed streamflow. Performance remains high even in log-
transformed variants of NSE and KGE, reflecting the model's ability to capture both high- and low-flow
regimes. Evaluation metrics assessing bias (Alpha-NSE) and variability (Beta-KGE) are close to 1.0 across
most regions, suggesting accurate representation of both amplitude and temporal dynamics of flow.
Crucially, this out-of-sample validation ensures that the model's skill reflects true generalization rather than
overfitting to known conditions—an essential distinction in the context of flood forecasting, where
performance metrics often involve comparisons across both training or calibration and test periods due to



limited data availability. By validating against held-out time periods, we demonstrate that Bakaano-Hydro
is not merely replicating known hydrological patterns, but can generalize in previously unseen contexts.
This enhances the robustness of our benchmarking framework, enabling a fair and unbiased comparison
with global systems such as GloFAS and the Google AI model. Figure A1 and A2 also compares the
hydrograph of Bakaano-Hydro predicted streamflow against GloFAS and Google AI model for the period
1982 – 2016, which covers both the training and testing period.

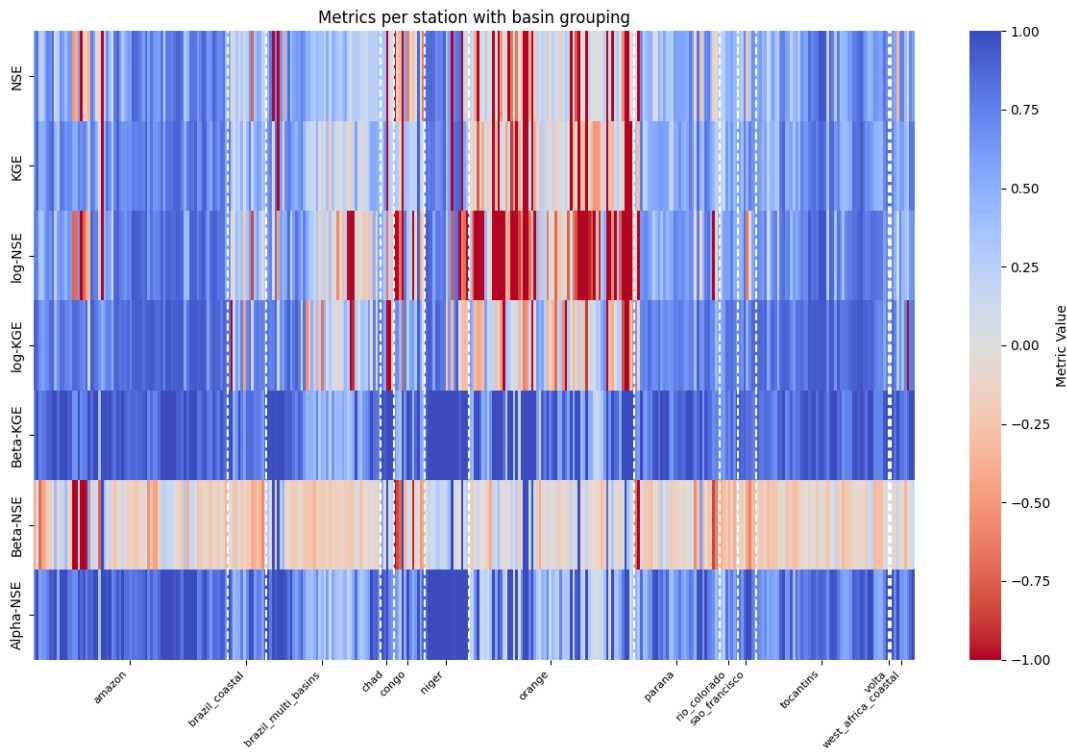

**Figure 2.** Out-of-sample evaluation (1982–1988) of Bakaano-Hydro performance across all 470 hydrological
stations used in the forecasting reliability assessment. Stations are grouped by basin, with dashed white
lines indicating basin boundaries. The heatmap displays multiple evaluation metrics, including Nash–
Sutcliffe Efficiency (NSE), Kling–Gupta Efficiency (KGE), log-NSE, and extended variants such as Beta-KGE,
Alpha-NSE, and Beta-NSE (Gupta et al., 2009). The color scale ranges from red (lower performance) to blue
(higher performance), enabling visual comparison of skill across metrics and basins.



## 3.2 Model intercomparison of flood forecasting reliability

Three metrics—probability of detection (POD), false alarm rate (FAR), and critical success index (CSI)—were employed to assess reliability of the proposed framework. POD quantifies the proportion of observed flood events accurately predicted by the models, FAR measures the rate of false positives (incorrect flood predictions), and CSI evaluates overall prediction accuracy by accounting for true positives and false alarms. Hindcast skill, assessed through CSI, POD, and FAR, is used as a measure of forecasting reliability. This hindcasting serves as a proxy for operational forecasting reliability, helping to identify strengths and limitations in each model's performance and informing future flood predictions. The use of hindcast skill as a measure of forecast reliability is a standard practice in climate and hydrological forecasting.

Similar retrospective evaluations have been conducted to compare GloFAS and Google's AI model (Nearing et al., 2024), with GloFAS itself relying on historical streamflow simulations to validate its forecast skill (Alfieri et al., 2013). CSI, POD and FAR values for flood events with 1-, 2-, 5-, and 10-year return periods and 0-, 1- and 2-day timing tolerance were calculated using simulated data from Bakaano-Hydro for the period 1982 to 2016. Timing tolerances are permissible temporal deviations between observed and simulated flood events. Performance was benchmarked against GloFAS and the Google AI model; notably, the Google AI model data used here were derived from a full model run encompassing all stations, rather than the cross-validation splits reported in Nearing et al. (2024). To ensure sufficient data for calculating return-period thresholds, we included both training and out-of-sample streamflow predictions from each model similar to Nearing et al. (2024) and Alfieri et al. (2013). A common subset of stations across the three models (470 stations in total) was selected for the calculation of these metrics. Return period thresholds were computed from observed data at each station and applied uniformly to the predictions of all models.

The results show that the median CSI scores of Bakaano-Hydro at a 0-day timing tolerance are comparable to or exceed those of GloFAS at 1-day and 2-day timing tolerance, and are comparable to Google AI model's 1-day timing tolerance CSI scores for every return period examined (Figure 3). The median scores of Bakaano-Hydro at 1-day timing tolerance are also higher than that of Google AI model at 2-day timing tolerance. These indicate substantial improvement in temporal precision of flood detection by Bakaano-Hydro. Figure 3 also reveals important differences in the reliability and variability of flood predictions from GloFAS, Google AI model, and Bakaano-Hydro, as illustrated by the shapes and ranges of their CSI distributions across Africa and South America. For GloFAS, the distribution skews markedly toward lower CSI values—particularly under stricter timing conditions (e.g., the 0-day timing tolerance) and at higher





**Figure 3.** Comparison of the CSI distribution for multiple return period flood events across three models—GloFAS, Google, and Bakaano-Hydro—under different timing tolerances (0-day, 1-day, and 2-day) in Africa and South America. Each panel presents half violin plot, boxplots and density plots to visualize the distribution of CSI values. The boxplot shows distribution quartiles and whiskers show the full range; the points show the distribution for the gauging stations and the half-violin plots.

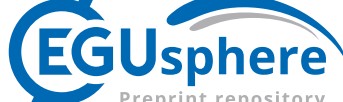

return periods—indicating consistently weak performance and a narrow range of outcomes. In contrast, the Google AI model displays broader distributions, generally clustering within a lower-to-mid CSI range. While it slightly outperforms GloFAS, its performance still declines considerably for rarer, more extreme flood events (e.g., from 1-year to 5-year return periods). Notably, Bakaano-Hydro exhibits a wider and more balanced distribution, with peaks shifting toward higher CSI values, reflecting a greater overall capacity for reliable flood prediction. Although Bakaano-Hydro's broader spread indicates some variability—meaning its performance can occasionally align with lower CSI values—the model consistently demonstrates higher median CSI scores than GloFAS and Google AI model. As timing tolerances expand from 0 to 2 days, all three models show higher CSI values, signaling that relaxing the timing tolerance can enhance apparent predictive skill. Statistical testing show that Bakaano-Hydro's superior CSI scores over GloFAS and Google AI model are significant in all return periods and timing tolerances. Comparing Bakaano-Hydro and GloFAS, p-values ranged from $5.8 \times 10^{-46}$ (Cohen's d = 0.64) to $7.0 \times 10^{-11}$ (Cohen's d = 0.19) and from $1.28 \times 10^{-32}$ (Cohen's d = 0.29) to $3.5 \times 10^{-4}$ (Cohen's d = 0.065) for Bakaano-Hydro and Google AI model.

Across all return periods and timing tolerances, the patterns in FAR (Figure 4) and POD (Figure 5) highlight important trade-offs that ultimately shape the CSI for each model. For all models, FAR decreases and POD increases as the timing tolerance widens (1–2 days). GloFAS consistently exhibits the highest FAR, particularly for shorter return periods and under strict (0-day) timing tolerances, indicating frequent over-prediction (i.e., many false alarms). Simultaneously, GloFAS shows the lowest POD, suggesting relatively few correct detections (hits) and a larger number of misses. This high FAR coupled with low detection directly hinders its CSI, which accounts for both misses and false alarms. By contrast, the Google AI model demonstrates moderate FAR levels and a broader range of POD outcomes. Nonetheless, the model's POD remains more variable and still includes lower-probability detections, especially for shorter return periods. Finally, Bakaano-Hydro attains the strongest performance across these metrics: it consistently maintains the lowest FAR and the highest POD. Even under strict 0-day timing tolerances, Bakaano-Hydro's false alarms remain minimal, and it achieves more hits relative to misses than the other models do.

Statistical testing show that Bakaano-Hydro's decrease in FAR and increase in POD scores over GloFAS and Google AI model are significant in all return periods and timing tolerances with p-values substantially lower than 0.05. Cohen's d analyses underscore substantial effect sizes favoring Bakaano-Hydro (ranging from 0.2 to 0.8), most pronounced in comparisons with GloFAS. For FAR, statistical testing of Bakaano-Hydro and GloFAS showed p-values ranging from $8.74 \times 10^{-35}$ (Cohen's d = -0.65) to $1.6 \times 10^{-9}$ (Cohen's d = -0.35) and p values ranging from $2.14 \times 10^{-23}$ (Cohen's d = -0.41) to $1.38 \times 10^{-5}$ (Cohen's d = -0.25) for Bakaano-Hydro



**Figure 4.** Comparison of the FAR distribution for multiple return period flood events across three models—GloFAS, Google, and Bakaano-Hydro—under different timing tolerance (0-day, 1-day, and 2-day) in Africa and South America. Each panel presents half violin plot, boxplots and density plots to visualize the distribution of FAR values. The boxplot shows distribution quartiles and whiskers show the full range; the points show the distribution for the gauging stations and the half-violin plots.





and Google AI model. For POD, statistical testing of Bakaano-Hydro and GloFAS showed p-values ranging
from 1.30 x 10$^{-55}$ (Cohen's d = 1.03) to 6.2 x 10$^{-17}$ (Cohen's d = 0.48)  and p-values ranging from 2.2 x 10$^{-46}$
(Cohen's d = 0.82) to 1.2 x 10$^{-10}$ (Cohen's d = 0.35)  for Bakaano-Hydro and Google AI model.

## 3.3 Spatial variability in forecasting reliability across hydroclimatic regions

Across diverse hydro-climatic contexts, the performance of GloFAS, the Google AI model, and Bakaano-
Hydro in predicting fluvial flooding varies notably, reflecting local hydrological complexities. Overall,
Bakaano-Hydro outperforms GloFAS and Google AI model across diverse hydroclimatic areas with the
improved performance in most basins statistically significant (Figures 6 and 7). Google AI model also shows
strong predictive skill across diverse basins, albeit under 2-day timing tolerance. GloFAS, being a physically-
based model, tends to perform well in larger, more predictable river basins but struggles in basins with
highly dynamic hydrological patterns.
Generally, across all models performance is lower in semi-arid to arid areas and higher in humid and
tropical areas. In semi-arid to arid regions—such as the Niger, Orange, Namibia coastal basins—
performance across all models is generally lower, with CSI values indicating difficulty in capturing the
intermittent and highly variable river flows, driven by extreme upstream rainfall that causes rapid river-
level rises, high transmission losses, and complex flood propagation. In these settings, GloFAS exhibits the
lowest CSI values mainly because of the highest false alarm rates, indicating an overestimation of flood
risk, while the Google AI model offers improved detection. Bakaano-Hydro outperforms GloFAS and Google
AI model in these settings with relatively higher CSI values. Figures 6 and 7 show the basins for which
differences in CSI and FAR were statistically significant.



**Figure 5**. Comparison of the POD distribution for multiple return period flood events across three models—
GloFAS, Google, and Bakaano-Hydro—under different timing tolerance (0-day, 1-day, and 2-day) in Africa
and South America. Each panel presents half violin plot, boxplots and density plots to visualize the
distribution of POD values. The boxplot shows distribution quartiles and whiskers show the full range; the
points show the distribution for the gauging stations and the half-violin plots.



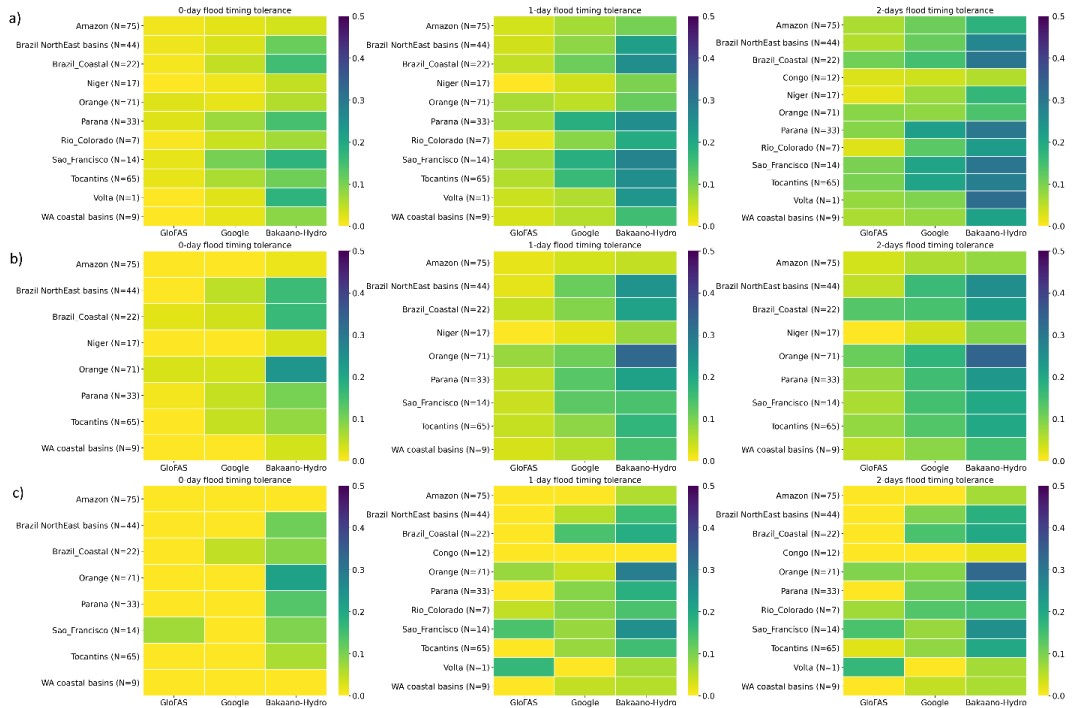

**Figure 6** Heatmaps of median CSI values for river basins across Africa and South America. a) shows heatmaps for floods with 1-year return period; b) floods with 2-year return periods; c) floods with 5-year return periods. For each plot only basins for which differences among the three models were statistically significant are shown. Brazil NorthEast basins, Brazil_Coastal and WA coastal basins are not officially recognized basin names but are used in this paper to refer to a group of small basins in Brazil and West Africa.

In tropical and equatorial climates, such West Africa coastal basins, the Congo, and the Amazon basins, prolonged and intense rainfall from monsoonal and convective systems introduces another layer of complexity. Spatially variable rainfall, seasonal soil saturation, and delayed flood wave propagation in large river systems often lead to overprediction by GloFAS and the Google AI model, as evidenced by higher FAR and moderate CSI. In temperate and subtropical basins such as the Paraná and Tocantins, models show consistent performance improvements with longer timing tolerances, indicating better predictability of hydrological events in these regions. GloFAS performs better in these regions compared to arid areas, reflecting its suitability for regions with relatively predictable seasonal cycles.

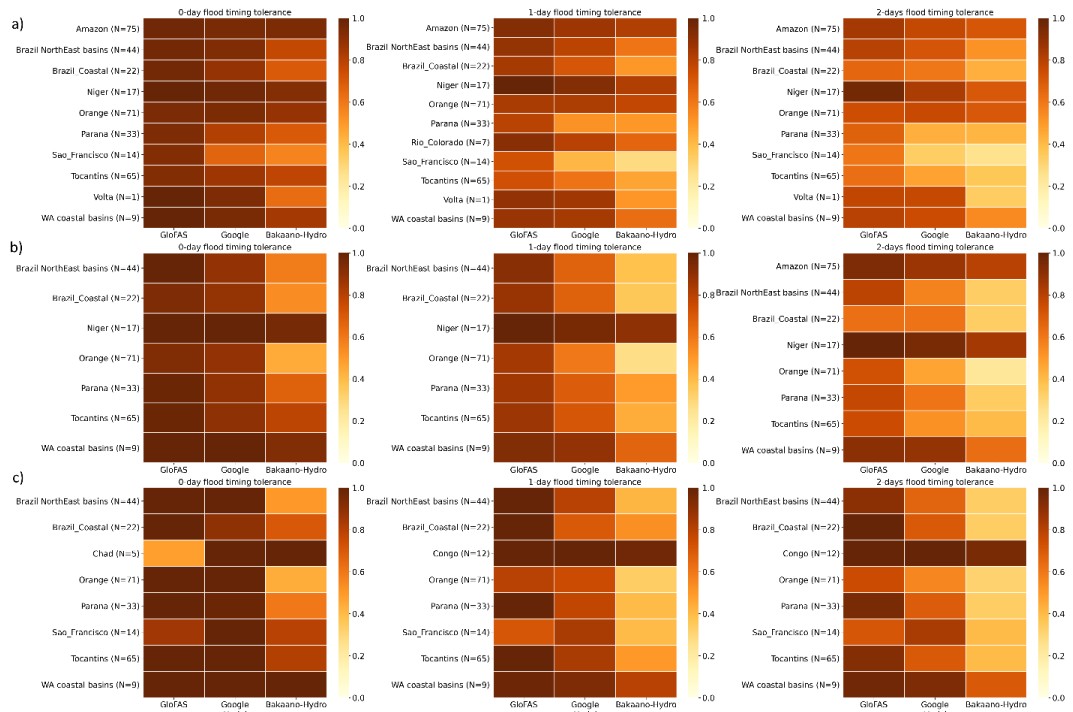

**Figure 7.** Heatmaps of median FAR values for river basins across Africa and South America. . a) shows heatmaps for floods with 1-year return period; b) floods with 2-year return periods; c) floods with 5-year return periods. For each plot, only basins for which differences among the three models were statistically significant are shown. Brazil NorthEast basins, Brazil_Coastal and WA coastal basins are not officially recognized basin names but are used in this paper to refer to a group of small basins in Brazil and West Africa.

## 4. Discussion

The results of this study demonstrate that the distributed hydrology-guided neural network framework significantly improves the reliability of flood forecasts in hydrologically diverse, data-scarce regions. Across a range of flood return periods, the model achieves higher CSI values and maintains lower FAR and higher probability of detection (POD) than both GloFAS and the Google AI model. Notably, Bakaano-Hydro outperforms GloFAS and Google AI model under precise timing constraints. Its performance at 0-day timing tolerance exceeds or matches GloFAS' performance at 2-day and Google AI model performance at 1-day timing tolerance. This enhanced temporal precision is critical for real-time flood early warning, especially in regions with high vulnerability and limited emergency response capacity. Importantly, in this paper, I



adopt a consistent benchmarking methodology by using observed streamflow data to define flood thresholds across all models—unlike model-specific thresholds that can artificially inflate or deflate performance metrics (e.g. Nearing et al., 2024). This evaluation approach enables a more objective and physically meaningful comparison, ensuring that improvements in forecast skill are attributable to model behavior rather than calibration biases.

Bakaano-Hydro contributes a new paradigm to the field of hydrological modeling by demonstrating that fully distributed, process-informed neural networks can reliably simulate streamflow and flood dynamics. It advances the state of the art in data-driven hydrological modelling by integrating process-based runoff generation and topographic flow routing with a deep learning architecture capable of generalizing across basins, climatic zones, and hydrological regimes. This overcomes existing trade-off between physical realism and data-driven scalability, offering a unified approach that performs well even in environments with limited or fragmented observational records. A key scientific insight is that Bakaano-Hydro achieves increased reliability despite being trained only on data from Africa and South America, regions typically underrepresented in global hydrological training datasets. In contrast, the Google AI model (Nearing et al., 2024) model draws on global data, including from data-rich regions such as Europe, North America, and Australia. That Bakaano-Hydro performs better or comparably—particularly in Africa and South America— highlights not only the framework's efficiency in learning from sparse, heterogeneous data but also the advantages of physically guided spatial representation over lumped learning schemes.

The development and evaluation of Bakaano-Hydro has far-reaching implications for climate resilience, disaster risk reduction, and sustainable development, particularly in the Global South. By offering a distributed, interpretable, and generalizable modeling framework, Bakaano-Hydro bridges a critical gap between global early warning systems and locally relevant flood risk information—a gap that has historically undermined emergency preparedness in data-scarce regions. For national governments and hydrometeorological agencies, Bakaano-Hydro offers a pathway toward strengthening domestic early warning capacity without requiring dense monitoring networks or extensive model calibration. Its modular and open-source architecture enables customization for local use cases—whether for national disaster response systems, basin authorities, or regional climate services. This opens up the possibility of transitioning from dependency on external, often coarse-resolution forecasts (like GloFAS or Google AI model) to context-specific, high-resolution models that are co-developed and owned by local institutions. For the humanitarian and development sector, Bakaano-Hydro provides a decision-support tool for anticipatory action, allowing earlier and more precise deployment of flood relief, social protection





schemes, and infrastructure safeguards. Its ability to generalize to ungauged basins also supports climate adaptation planning and nature-based solutions, such as ecosystem restoration in floodplains and watershed management. At the global scale, Bakaano-Hydro advances the science-policy interface by demonstrating that physically informed machine learning models can be both scalable and regionally accurate—paving the way for hybrid forecasting systems that serve diverse geographies, socio-economic contexts, and governance capacities. From a scientific standpoint, the framework opens new pathways for interpretable machine learning in hydrology. By using runoff as an intermediate, physically meaningful variable, Bakaano-Hydro avoids the black-box pitfalls of purely statistical models and supports diagnostic evaluation of model behavior. The architecture also enables flexible experimentation with modular components—allowing researchers to test how different land surface models or routing schemes influence predictive performance across space and time. Operationally, Bakaano-Hydro offers a practical and scalable solution for flood forecasting in regions with limited data infrastructure, where conventional physically based models struggle to calibrate, and global lumped models fail to resolve local dynamics.

A primary limitation of the current study is its use of historical observed discharge records rather than true forecast or reforecast datasets. While this retrospective analysis enables rigorous comparison across models and return periods, it does not fully capture the operational uncertainties present in real-time forecasting—such as delays in data assimilation, atmospheric forecast errors, or recent changes in land use and hydraulic infrastructure. Moreover, under ongoing climate change, stationarity assumptions may become less valid, potentially limiting the relevance of hindcasting performance as a predictor of future skill. Nonetheless, the long-term, multi-basin dataset used in this study spans a wide range of hydrological conditions and extreme events, offering a robust foundation for model validation. A model that performs reliably across these historical extremes is well-positioned to adapt to future conditions when coupled with evolving inputs such as satellite rainfall data and climate forecast ensembles.

## 5. Conclusion

This study demonstrates the potential of a distributed, hydrology-guided neural network framework that integrates process-based runoff generation, topographic flow routing, and temporal convolutional networks to improve flood forecasting reliability in data-scarce regions. The framework—Bakaano-Hydro—was benchmarked against GloFAS and the Google AI model using retrospective simulations across 470 gauging stations in Africa and South America. Results show that Bakaano-Hydro consistently achieves higher forecasting reliability across multiple flood return periods and timing tolerances. Statistically



significant improvements in critical reliability metrics, including the critical success index (CSI), probability
of detection (POD), and false alarm rate (FAR), indicate the model's robustness in capturing both the
occurrence and timing of flood events.
Importantly, Bakaano-Hydro advances the state of data-driven hydrological modeling by embedding
physically meaningful processes within a neural network architecture. This hybrid approach addresses key
limitations of lumped data-driven models and calibration-intensive physically based systems, offering a
generalizable and interpretable alternative suited to diverse hydroclimatic contexts. The ability of the
framework to generalize from sparse training data, while maintaining physically plausible representations
of runoff and flow routing, highlights its potential to support robust hydrological analysis in observationally
limited settings. Overall, Bakaano-Hydro contributes a methodologically rigorous and scientifically
grounded step forward in the development of hybrid flood forecasting models.

## Code availability

Source codes for Bakaano-Hydro are available at https://github.com/confidence-duku/bakaano-hydro.
The neural network architecture in this version was adapted for this study and can be found at
https://doi.org/10.5281/zenodo.15322955. All codes for Bakaano-Hydro evaluation and return period
estimation can be found at
https://github.com/google-research-datasets/global_streamflow_model_paper.

## Data availability

Daily streamflow data for 1982 – 2016 simulated for hydrological stations across Africa and South America
as well as validation and CSI, POD, FAR results are available at https://doi.org/10.5281/zenodo.15322955.
GloFAS and Google AI model benchmark data are available at https://doi.org/10.5281/zenodo.10397664

## Competing interests

The authors declare that they have no conflict of interest.





**Appendix A**

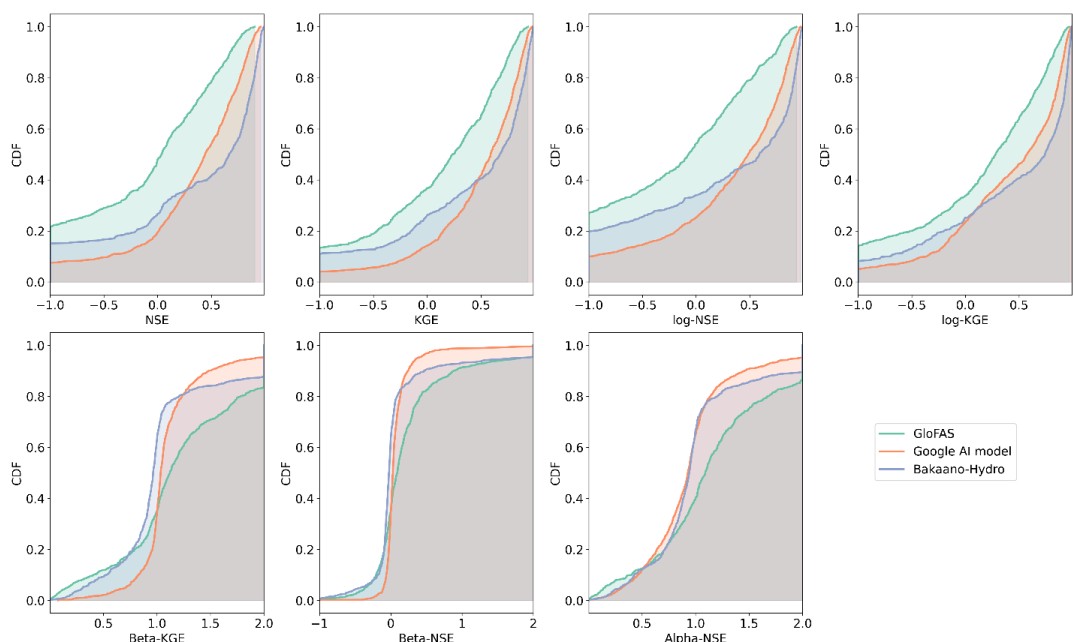

**Figure A1.** Comparison of cumulative distribution functions (CDFs) of hydrological performance metrics
across values for the three models - GloFAS, Google AI model, and Bakaano-Hydro - for the period 1982
- 2016. Metrics include Nash-Sutcliffe Efficiency (NSE), Kling-Gupta Efficiency (KGE), log-NSE and
additional variants such as Beta-KGE, Alpha-NSE and Beta-NSE



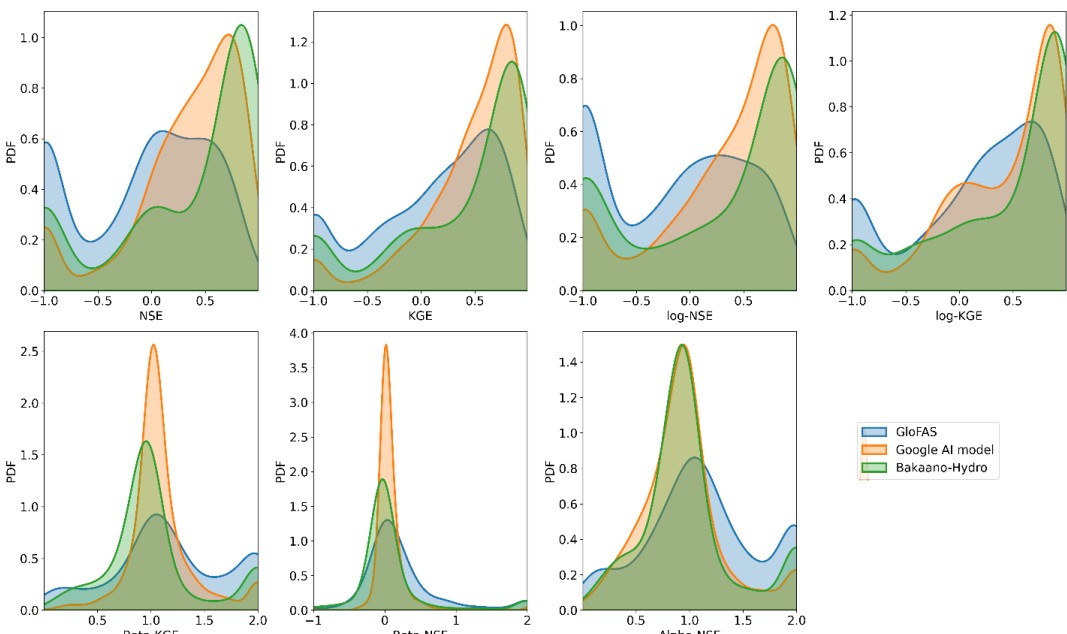

Figure A2. Comparison of probability density functions (PDFs) of hydrological performance metrics across
values for the three models -  GloFAS, Google AI model, and Bakaano-Hydro -  for the period 1982 - 2016.
Metrics include Nash-Sutcliffe Efficiency (NSE), Kling-Gupta Efficiency (KGE), log-NSE and additional
variants such as Beta-KGE, Alpha-NSE and Beta-NSE

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

Disaster Risk Reduction.

