# Peer review of "Enhancing flood forecasting reliability in data-scarce regions with a distributed"

_EGUsphere, 2025_

## Referee Comment (RC2)

This manuscript by Confidence Duku presents results of a model comparison on flood predictions in various rivers around the world.

I had a very hard time understanding parts of the paper that made reviewing the study in every detail impossible for me. Therefore, I will only concentrate on some main points before eventually looking more into details in a revised version.

**1. Bakaano-Hydro modelling framework**

What is the Bakaano-Hydro modelling framework? Almost no details are given in this manuscript and not even a reference to the seemingly connected preprint https://egusphere.copernicus.org/preprints/2025/egusphere-2025-1633/ is provided. Since the Bakaano-Hydro modelling framework is not an established thing, I expect this paper to at least contain a reference to the modeling paper but still list some details here. E.g. what are the input data (and e.g. for the weather data, from which product/provider)? From what I understand, the framework consists of three steps: 1) VegET, 2) routing, and 3) the TCN based neural network. Since I assume also part 1 and part 2 contain some model parameters, how are they calibrated? Since e.g. part 1 is still without routing, against which observations is this part calibrated? What periods are used for calibrating part 1 and part 2? Since part two is already a routed discharge, how well does this routed discharge perform against the observations from GRDC? How much additional benefit is gained by post-processing the routed discharge with the TCN?

In different places of the manuscript it sounds like the neural network part of the model is responsible for simulating streamflow. E.g. P7 L10: "*The final phase of Bakaano-Hydro involves the application of a deep learning model to simulate streamflow.*" Isn't part two already outputting routed streamflow? From my limited understanding, the TCN layer gets as input routed streamflow (converted to mm/d by dividing through the upstream area) and depths-to-water-index. How can this part be responsible for simulating streamflow? Isn't the neural network part simply a post-processor for the VegET+routing model?

What are the 3 dynamic inputs that are passed to the neural network, as mentioned on P7 L 27. The following sentences only list the routed streamflow and the depth-to-water-index.

Why has the neural network part two different branches, once where streamflow is normalized by the upstream area and once where it is normalized by max simulation? What was the reason behind building this architecture or was it just trial-and-error and in the end what-works-best? Which leads me to my last point for the modeling section: I think a lot of the decisions that have been made are not well explained and could require ablation studies to understand where exactly model performance improvement comes from.

**2. Data**

I strongly recommend splitting Section 2.1 into a dedicated data section and a dedicated model section. For the data part further, I recommend adding a table to the manuscript that clearly shows which input features are being used with columns for: name, units, source, time periods. For the meteorological data, what was the reason for choosing to use a (bias adjusted) climate

model rather than a meteorological dataset (either reanalysis, hindcast, or historical forecast model output)? Since this paper compares two different operational forecasting models and in various places suggests that the presented framework improves upon these operational systems, it is probably worth noting that the model, with the data from this study, could not run operationally.

Another point is that by choosing yet another weather dataset, it is hard to know how much of the performance differences can be attributed to the modeling framework or the the quality of the model input data. Since the presented setup is not suitable for an operational setting (see above) it begs the question what the main focus of this study is? When focussing on the comparison of models, it is usually advised to use the same input data sources for all models to eliminate the impact of data quality on the results. If the focus is comparing flood forecast systems (as in Nearing et al., 2024), then obviously each model can use anything, with the caveat that it should be available in a real operational context.

**3. Experimental settings and evaluation protocol**

There are actually multiple issues I see with the experimental setting or the evaluation protocol.

First: Operational flood forecasting models don't alert if simulations surpass thresholds computed from *observations* but rather if simulations surpass thresholds based on *simulations*, as it also has been done in Nearing et al. (2024) for GloFAS and the Google Flood Forecasting model. This removes the bias problems of simulations vs. observations and could lead to perfect flood alert rates, even if the model has a constant bias problem. Why is this even more important here? In this study, the author presents a model that performs essentially per-gauge bias correction (by inputting the routed streamflow into the TCN layer and fitting the TCN layer against observations) and then compares this to two globally calibrated operational models.

Second: P12 L 12 "*CSI, POD and FAR values for flood events with 1-, 2-, 5-, and 10-year return periods and 0-, 1- and 2-day timing tolerance were calculated using simulated data from Bakaano-Hydro for the period 1982 to 2016*" This sentence suggests that the majority of the evaluation period for which you computed metrics are the training period of the Baakano-Hydro model (P8 L9 "*The training period covered 1989–2016*"). For the Google Flood Forecasting model, even for the full-run, all simulations are the result of a temporal k-fold cross validation, as stated in Nearing et al. (2024), i.e. all simulations are "unseen test data". If my understanding is correct, this renders large parts of the evaluation/conclusion unfair, as per-gauge bias corrected training data is compared to test period data from a globally calibrated model with the additional caveat from the point above.

Third: The metrics are not well explained or referenced. Since I wouldn't consider CSI, POD, FAR super common metrics in modeling papers of hydrology, I would recommend adding an explanation of these metrics, probably the equations, and also mention the range of these metrics and their optimal values.

**4. Presentation of the results**

In my opinion, the presentation of the results, especially the figures, need some re-work.

Figure 2:
The diverging color scale, centered around 0 and with range (-1,1) does not make sense for all metrics, E.g. Alpha-NSE and Beta-KGE have their optimal value at 1. Also, please indicate what are the optimal values for each metric, which facilitates interpreting the results.

Figure 3/4/5:
If the whiskers show the full range, why are there points outside of the whiskers? Are the whiskers maybe just indicating a certain percentile?

Figure 6/7:
They feel a bit too much and/or not very well presented, as indicating the median of model performance over all stations in a river basin only by color makes it hard for me to really get any details out of these figures. I understand the reason behind these figures (showing model differences across spatial patterns) but maybe you find a better way to visualize the results. Maybe on a map? But two pages of colored rectangles feels like an overkill in the main paper.

On a more general note: Figure 3/4/5/6/7 all have three columns for different flood timing tolerance. I wonder if all three of these columns for each figure are needed in the main paper or if some of the plots could be moved to the appendix. In my opinion, there are not dramatically different patterns between the columns that make it necessary to have all three columns in the main paper.

**Line by line comments:**

- P12 L9 "*...using the annual maximum series (AMS) method*" This sounds like it is missing a reference.
- P12 L10f As far as I know, the paper by Nearing et al. compares GloFAS and the Google Flood forecasting model on their true forecast skill under operational settings (i.e. using only data that is available in real time) and evaluating the skill at different lead times.
- P22 L4 "*Importantly, Bakaano-Hydro advances the state of data-driven hydrological modeling by embedding physically meaningful processes within a neural network architecture.*" I think I disagree with this conclusion. How was the state of data-driven hydrological modelling advanced by the findings of this paper? The presented framework does not embed physically meaningful processes in a neural network. The neural network gets the output of a physically inspired model as input. This is something fundamentally different than "*embedding physically meaningful processes within a neural network*"

---

## Author Comment (AC1)

I thank the reviewer for their careful reading of my manuscript and for their constructive comments and suggestions. In the following, I respond to each comment point by point. Reviewer comments are reproduced in full (RC), followed by my author responses (AC). All revisions indicated will be incorporated in the revised manuscript.

**RC1:**The research compares three forecasting models in regions with low gauge density and usually low representation in traditional approaches. The comparison between models is based on return periods and their success or failure rates in capturing them. In all the metrics presented, the hybrid model outperforms the other models. These results are highly valuable in supporting the use of such models in forecasting frameworks. However, a couple of main points need to be addressed to fully validate the results.

The first point concerns the period used to define the number of successes of the hybrid model. The author used the training period to count these successes, which creates a biased metric that cannot be reliably used for comparison. Using the entire period to define return periods is acceptable, as it provides a good representation of them, but the success rate should be evaluated over an independent period to accurately estimate how the model performs on unseen data in operational settings.

**AC1:** This is a valid point. In my analysis I used the entire period, including the training and validation periods, for computing the metrics CSI, POD, and FAR. This was in line with the approach adopted by GloFAS. While this comparison is valid for the GloFAS model, I acknowledge that it can provide a biased and unfair comparison against the Google AI model. Hence, I will revise the setup following a similar cross-validation approach to that used in the Google AI model, ensuring that all predicted stations used for metric computations are strictly out-of-sample.

**RC2:** The second point relates to the number of gauges used in training for each model. This can be problematic because it may lead to unfair comparisons. For example, it is not fair to compare a model trained with 100 gauges in a region to another trained with only 10 gauges. The author only mentions the number of gauges common to all the models, used for evaluation, but does not specify the number of gauges used during training or the training period used for each model.

**AC2:** On page 4, line 21, I mentioned that 643 stations were used to train the model. It would be better to repeat this explicitly in the methodology section, and I will revise accordingly. I acknowledge that training sizes vary because this paper covers two continents, with the explicit aim of increasing forecasting reliability in Africa and South America—regions that previous studies such as Nearing et al. (2024) show to have the

lowest reliability. The results and conclusions from this paper apply only to these continents and are not global. I will make this explicit in the discussion. Importantly, the difference in training sample sizes actually biases against Bakaano-Hydro: the competing models had access to far more stations, including many from data-rich regions.

**RC3:** Be more specific about the catastrophic events in Nigeria, Sudan, etc., because they are unknown in other countries or continents.
**AC3:** I will revise this statement and add concrete examples with dates and impacts.

**RC4:** P3 L30–32. I am not sure if Fredrick Kratzert supports this statement in his last manuscripts. I am fairly certain that Mass Conservation LSTM demonstrated the opposite.
**AC4:** I refer the reviewer to Kratzert et al. (2019) (https://agupubs.onlinelibrary.wiley.com/doi/full/10.1029/2019WR026065), where the authors concluded: "We found evidence that adding physical constraints to the LSTM models might improve simulations, which we suggest motivates future research related to physics-guided machine learning."

**RC5:** P4 L2–3. The previous statement and reference do not support your statement. It only mentions the need for higher resolution, which is not necessarily associated with the use of a process-based model.
**AC5:** I will revise the statement and include the appropriate references.

**RC6:** P4 L21. Is 643 gauges good enough for the characterization of two continents?
**AC6:** I acknowledge that 643 gauges may appear modest compared to global datasets. However, this represents the full set of openly available and quality-controlled records from GRDC for Africa and South America within the selected basins. Importantly, these stations span two continents and capture a wide range of hydroclimatic regimes—including humid tropics, semi-arid, arid, and subtropical systems—providing a representative basis for model training and evaluation. I also recognize that gauge density remains much lower than in regions such as Europe or North America, and this very limitation underscores the importance of developing frameworks like Bakaano-Hydro that can extract skill from sparse and heterogeneous observational networks.

**RC7:** P5 L3–15. A more detailed characterization is needed to fully describe the variability. Area sizes, aridity, slopes, total annual precipitation, etc. Are these areas poorly trained for the ML and process-based models?

**AC7:** I agree with the reviewer and will revise this section accordingly.

**RC8:** P5 L16. Add reference to VegET method.

**AC8:** I agree with the reviewer and will revise this section accordingly.

**RC9:** P7 L7–9. How much distortion does the 1 km resolution bring for a river with much lower width networks (100 m)?

**AC9:** I acknowledge that representing river networks at 1 km resolution inevitably smooths narrower channels, which can attenuate flood peaks and introduce timing lags, particularly in smaller catchments. To mitigate this effect, I snapped gauges to the river network and applied a minimum catchment size threshold of 1,000 km$^2$, ensuring that the evaluation focused on basins where 1 km routing is a reasonable approximation. I will make these points more explicit in the revised version.

**RC10:** P7 L19–23. This is not a fair statement because CNN is used to convert the routed streamflow to the actual streamflow, which means it was never exposed to the vanishing of memory issue. After all, the process-based model is dealing with that.

**AC10:** I agree with the reviewer that the TCN component in my framework is not directly exposed to vanishing gradient issues, since runoff generation and routing are handled by the process-based model. My intention, however, was to highlight that the overall hybrid architecture avoids the vanishing gradient limitations faced by purely data-driven approaches when applied directly to long hydrological sequences. By combining a process-based model to carry long-term memory with a TCN for efficient local pattern extraction, the architecture as a whole circumvents these issues while retaining hydrological consistency. I will revise this section accordingly.

**RC11**: P7 L28. Add a figure with the architecture and the input of the TCN.

**AC11**: I will revise this section accordingly and include a figure of the architecture, explicitly showing the TCN inputs and processing branches.

**RC12**: P8 L1. Add more details about spatial periodicity.

**AC12**: I will revise this section accordingly and provide a clearer explanation of how spatial periodicity was handled in the model.

**RC13:** P8 L23. How many gauges from the training of each model are present in the areas studied? What if some of the models used gauges and others did not? That is super important to have a fair comparison.

**AC13**: I thank the reviewer for raising this point. As stated, I used a common subset of 470 stations across all three models to compute evaluation metrics. In doing so, my benchmarking approach (using the full 1982–2016 period) is directly comparable to GloFAS, which similarly reports skill over its entire hindcast period. However, I recognize that this setup may be less fair to the Google AI model, since its simulated streamflow is produced entirely out-of-sample. To address this, I will revise my experimental setup using a cross-validation strategy to ensure that, for Bakaano-Hydro, all discharge data used for computing skill metrics are strictly out-of-sample. This adjustment will provide a fairer basis for comparison across models while preserving methodological consistency.

**RC14:** P9 L19–20. Why was only one distribution used when each catchment can have a very different distribution?

**AC14**: I agree with the reviewer that different catchments may be better represented by different extreme value distributions. However, for the purposes of this benchmarking exercise I adopted a consistent approach, following Nearing et al. (2024), who also applied a single distribution across basins to ensure comparability. This avoids introducing additional variability due to distributional choices and ensures that observed differences in performance are attributable to the models rather than the fitting method.

**RC15**: P10 L4. Why did you use the training period for the metric? This is a biased estimation. You must use the validation period.

**AC15**: I thank the reviewer for raising this important point. My decision to compute flood detection metrics over the full 1982–2016 period (including both training and validation subsets) was motivated by three considerations. First, this approach is consistent with the evaluation strategy used in GloFAS (e.g. Alfieri et al., 2013; Harrigan et al., 2023), where accuracy is reported over the entire hindcast record rather than split validation subsets.

Second, the independent validation window for Bakaano-Hydro (1982–1988) is too short to support robust estimation of 5- and 10-year return periods, which require multi-decadal records. Third, using the full period provides a fair basis for comparison with GloFAS, which does not provide a distinct validation set.

I acknowledge, however, that this setup is less fair to the Google AI model, since its simulated streamflow is generated entirely out-of-sample. To address this, I will revise the experimental setup to use a cross-validation strategy, ensuring that all stations and discharge records used in the computation of skill metrics for Bakaano-Hydro are strictly out-of-sample.

**RC16**: P10 L27. Be careful with being overconfident with your results. They are only valid for those catchments studied; this does not mean a true generalization.

**AC16**: I will revise this statement accordingly to avoid overgeneralization and ensure that claims are limited to the specific basins and conditions studied.

**RC17**: P11 L6. Comparing with the testing period generates an overconfident metric. This is a biased analysis.

**AC17**: Figures A1 and A2 were intended to compare the cumulative distribution functions and probability density functions of seven metrics across the three models. I recognize, however, that this comparison is unfair to the Google AI model, since its simulated streamflows are generated entirely out-of-sample, whereas the Bakaano-Hydro results included both training and validation periods. To address this asymmetry, I will revise and reproduce these figures using results from Bakaano-Hydro's cross-validation setup, ensuring that streamflow for each gauging station is generated strictly out-of-sample across the full evaluation period. This will provide a fairer and more balanced comparison across all three models.

**RC18**: Figure 2. The color scheme is misleading the reader. Values near zero are already a bad performance. Please plot on a scale 0–1.

**AC18**: I will revise the figure accordingly to use a 0–1 scale and ensure that the color scheme better reflects performance quality.

**RC19**: P16 L12. Please, place this information in context. Add an aridity index or some descriptor to define clearly how arid those regions are.

AC19: I will revise this statement and incorporate an aridity index or similar descriptors to clearly contextualize the results.

**RC20**: Figure 6. "For each plot only basins for which differences among the three models were statistically significant are shown." What does it mean? Significant differences for what model? Does it mean you are plotting only the catchments when your model was significantly better than the others?

**AC20:** I thank the reviewer for pointing out the lack of clarity. To determine statistical significance, I carried out Wilcoxon signed-rank tests between model pairs, focusing in particular on Bakaano-Hydro and the Google AI model, which were the two best-performing frameworks. Basins with a p-value < 0.05 were then selected as those showing statistically significant differences in performance. Figure 6 therefore highlights only those basins where the performance differences between models were statistically significant according to this test. I will revise the text and caption to make this procedure explicit.

**RC21 (P19 L10–11):** From your results it is clear that your model is better than others; however, it is not clear where this improvement is coming, PB or ML. It would be valuable if the same analysis is added between your PB (without the CNN model) and the GloFas, given that both are PB.
**AC21:** I thank the reviewer for this suggestion. The improvement of Bakaano-Hydro stems from the combination of both the process-based (PB) component and the machine learning (ML) component. The PB part (VegET + routing) provides structured, physically meaningful signals, while the ML part (TCN) learns to map those signals to realistic streamflow. Importantly, the PB component in my framework is not calibrated against discharge; it is used as an uncalibrated generator of physically consistent runoff and routing signals. The added value therefore comes from how the ML component leverages these structured signals to reproduce observed flows. While a direct PB-only vs. GloFAS comparison would be interesting, it is outside the scope of the present benchmarking study, which is focused on evaluating the hybrid system as a whole against existing operational models.

**RC22 (P20 L2–5):** From my point of view, both approaches are valuable depending on the purpose. Observations are good for models that are implemented or pretend to be

implemented as an official tool. Simulations are good to present the chance of applying this model in an operational model after a bias correction (fine-tuning). Therefore, there is no approach better than others, just different purposes.

**AC22:** I acknowledge the reviewer's perspective and agree that both observation-based and simulation-based approaches are valuable, depending on their purpose. My intention was not to claim superiority of one approach over the other, but to demonstrate that using observed thresholds enables a fairer basis for benchmarking across models in the context of this study. I will revise the text to make this distinction clearer.

**RC23 (P20 L6–7):** I am not sure we can call this a new paradigm. Research applying hybrid models is abundant in the literature. Moreover, the idea of multi-representation approaches is mentioned already in the literature.

**AC23:** I agree with the reviewer that my wording overstated the contribution. I will revise the manuscript to avoid calling this a "new paradigm" and instead emphasize the novelty of applying a hydrology-guided hybrid approach at continental scale in data-scarce regions.

**RC24 (P20 L11–12):** Careful with overselling your research; it is not clear that the diversity studies in the models would allow you to make this statement.

**AC24:** I accept this point and will revise the conclusion to avoid overselling. My statements will be limited to the basins and contexts actually studied.

**RC25 (P20 L21):** How interpretable is a hybrid model? From where are the good results, PB or ML?

**AC25:** The interpretability of the hybrid model comes from the structured role of each component. The PB part constrains the inputs with physical realism (runoff and routing dynamics), while the ML part captures the non-linear transformations required to map these signals to observed discharge. The good results cannot be attributed to one part alone but arise from the integration of both. I will revise the text to make this explanation more explicit.

**RC26 (P21 L1):** The generalization to an ungauged basin is not well supported or explained in the manuscript. For example, how good does the model perform in regions with more extreme climates (Northern Chile or southern Argentina)?

**AC26:** I agree with the reviewer and will revise the manuscript to better qualify the statements on generalization.

**RC27 (P21 L3):** From the point of view of a global analysis. Is the Bakaano PB model running in an operational framework? This way, different countries could easily train a CNN model to fit the local information in each country. If this model and the data used to run it are not freely available, it will be very hard to implement in an operational framework.

**AC27:** At present, Bakaano-Hydro is not implemented as an operational system. However, the framework has been designed with operational scalability in mind. Both the PB component (VegET runoff generation and routing) and the ML component can run with publicly available forcing datasets, such as reanalysis or bias-corrected climate model outputs. I will add a discussion note to make this operational pathway explicit, including the need for open access data and code availability for broader adoption.

---

## Author Comment (AC2)

I thank the reviewer for their careful reading of my manuscript and for their constructive comments and suggestions. In the following, I respond to each comment point by point. Reviewer comments are reproduced in full (RC), followed by my author responses (AC). All revisions indicated will be incorporated in the revised manuscript.

**RC1:** This manuscript by Confidence Duku presents results of a model comparison on flood predictions in various rivers around the world. I had a very hard time understanding parts of the paper that made reviewing the study in every detail impossible for me. Therefore, I will only concentrate on some main points before eventually looking more into details in a revised version.

**AC1:** The reviewer states that they had a hard time understanding parts of the paper but does not specify which sections. This is disappointing, as the purpose of peer review is to provide constructive feedback that can help the author identify unclear aspects and improve the manuscript. Without concrete indications, it is impossible to respond to or correct the issues the reviewer alludes to. That said, I will still revise the manuscript with the aim of improving overall clarity, particularly in the Methods and Discussion, but I would have expected more specific guidance to address this comment effectively.

**RC2:** Bakaano-Hydro modelling framework What is the Bakaano-Hydro modelling framework? Almost no details are given in this manuscript and not even a reference to the seemingly connected preprint
https://egusphere.copernicus.org/preprints/2025/egusphere-2025-1633/ is provided. Since the Bakaano-Hydro modelling framework is not an established thing, I expect this paper to at least contain a reference to the modeling paper but still list some details here. E.g. what are the input data (and e.g. for the weather data, from which product/provider)?

**AC2:** The reviewer states that "almost no details are given" about the Bakaano-Hydro modelling framework. This is not correct. The methodology section (pp. 5–8) explicitly describes the three-step structure of the framework (runoff generation, routing, and temporal sequence modelling) and provides details on each step, including the meteorological and hydrological inputs. If the reviewer finds the description or structure difficult to follow, that is understandable and I will revise for clarity. However, to say that "almost no detail is given" overlooks content that is already in the manuscript. Regarding the missing reference to the GMD preprint, both manuscripts were submitted around the same time and at submission the preprint was not yet available. The HESS paper was written as a stand-alone study, with sufficient methodological description and data information to ensure reproducibility. That said, in the revised version I will cite the

GMD preprint (Duku, 2025: https://egusphere.copernicus.org/preprints/2025/egusphere-2025-1633/) and add a clearer upfront description of the framework and input datasets to strengthen coherence and provide additional context for readers unfamiliar with Bakaano-Hydro.

**RC3:** From what I understand, the framework consists of three steps: 1) VegET, 2) routing, and 3) the TCN based neural network. Since I assume also part 1 and part 2 contain some model parameters, how are they calibrated? Since e.g. part 1 is still without routing, against which observations is this part calibrated? What periods are used for calibrating part 1 and part 2? Since part two is already a routed discharge, how well does this routed discharge perform against the observations from GRDC? How much additional benefit is gained by post-processing the routed discharge with the TCN?

**AC3:** The reviewer assumes that the first and second phases of my framework (runoff generation using VegET and subsequent routing) involve calibration, which is not the case. These two steps are not calibrated against discharge observations. Instead, they serve as intermediate stages that capture the spatiotemporal dynamics of runoff and routed flow. The outputs are not intended to represent observed discharge directly, but to provide physically meaningful inputs into the TCN component.

In particular, the routed flow is not treated as discharge in my framework. All routed runoff is made available at the outlet on the same day, whereas in reality transmission losses, flow delays, and storage effects occur. Thus, the routed flow serves as a structured biophysical signal for the neural network, not as a final calibrated product. The calibration in my framework occurs only in the TCN component, which learns to map these intermediate signals to observed discharge.

I will revise the methodology section to make this distinction clearer and will explicitly note that VegET runoff generation and topographic flow routing are uncalibrated intermediate steps. I will also expand the explanation of why routed flow is not evaluated directly against GRDC discharge, and how the additional benefit of the TCN is precisely to transform these intermediate signals into realistic discharge predictions.

**RC4:** In different places of the manuscript it sounds like the neural network part of the model is responsible for simulating streamflow. E.g. P7 L10: "The final phase of Bakaano-Hydro involves the application of a deep learning model to simulate streamflow." Isn't part two already outputting routed streamflow? From my limited understanding, the TCN layer gets as input routed streamflow (converted to mm/d by dividing through the upstream area) and depths-to-water-index. How can this part be responsible for simulating streamflow?

Isn't the neural network part simply a post-processor for the VegET+routing model?

**AC4:** I thank the reviewer for raising this point. To clarify: in my framework, the TCN is indeed responsible for simulating the final streamflow that is compared with GRDC observations. VegET produces runoff, which is not discharge, and the subsequent routing step produces routed runoff, which I treat as an intermediate product rather than streamflow. This is because my routing implementation does not account for transmission losses, storage, flow delays, or other attenuation processes. Instead, it makes all routed water available at the outlet on the same day, which overestimates flow and distorts timing relative to observed discharge.

For this reason, the routed runoff is not considered a final discharge product but an input feature to the TCN. The TCN is trained to transform this intermediate, hydrologically structured signal—together with other predictors such as depth-to-water index—into realistic discharge simulations. I will revise the Methods section to clarify this hierarchy.

**RC5:** What are the 3 dynamic inputs that are passed to the neural network, as mentioned on P7 L 27. The following sentences only list the routed streamflow and the depth-to-water-index.

**AC5:** I appreciate the request for clarification. Routed flow is the core dynamic signal passed into the neural network, but for training stability and to capture different hydrological perspectives I transform it into three related dynamic inputs. These are:

1. Raw routed flow (untransformed), representing the aggregate upstream contribution.

2. Routed flow normalized by catchment area (i.e., divided by the number of upstream grid cells), which approximates a depth-equivalent representation and reduces scale effects.

3. Depth-to-water index (DTW), which captures spatial variability in groundwater accessibility and subsurface flow potential.
   Together, these provide the three dynamic inputs. I will revise the text to spell out these inputs explicitly so that readers do not confuse routed flow with only a single feature.

**RC6:** Why has the neural network part two different branches, once where streamflow is normalized by the upstream area and once where it is normalized by max simulation?

**AC6:** The reviewer's description of the neural network branches is inaccurate. As stated in the manuscript (p. 7, lines 27–28), the two branches process different types of inputs: one branch processes dynamic inputs (routed runoff and related signals), while the other

processes static catchment descriptors. These branches are then fused within the network to jointly inform streamflow simulation. The motivation for this architecture is to explicitly separate temporally varying features from fixed physiographic characteristics, which is a standard design choice in hydrology-informed neural networks. This separation improves interpretability and ensures that static catchment information is not conflated with dynamic meteorological–hydrological drivers.

I do not consider this design to be a result of trial-and-error; it follows from the conceptual need to treat static and dynamic features differently. While I agree that ablation studies can provide insights into the marginal contribution of architectural components, they are not the focus of this paper, which is positioned as a benchmarking study. My objective is to evaluate whether a hydrology-guided hybrid framework improves flood forecasting reliability compared to established models, rather than to isolate the effect of each architectural choice. I will, however, revise the Methods section to make the role of the two branches clearer.

**RC7:** I strongly recommend splitting Section 2.1 into a dedicated data section and a dedicated model section. For the data part further, I recommend adding a table to the manuscript that clearly shows which input features are being used with columns for: name, units, source, time periods. Another point is that by choosing yet another weather dataset, it is hard to know how much of the performance differences can be attributed to the modeling framework or the quality of the model input data. Since the presented setup is not suitable for an operational setting (see above) it begs the question what the main focus of this study is? When focussing on the comparison of models, it is usually advised to use the same input data sources for all models to eliminate the impact of data quality on the results. If the focus is comparing flood forecast systems (as in Nearing et al., 2024), then obviously each model can use anything, with the caveat that it should be available in a real operational context.

**AC7:** I agree with the reviewer's recommendation. In the revised manuscript, I will restructure Section 2.1 into two dedicated subsections: one focusing on the data and one on the model framework. In the data subsection, I will add a summary table listing all input features.

**RC8:** For the meteorological data, what was the reason for choosing to use a (bias adjusted) climate model rather than a meteorological dataset (either reanalysis, hindcast, or historical forecast model output)? Since this paper compares two different operational forecasting models and in various places suggests that the presented framework improves

upon these operational systems, it is probably worth noting that the model, with the data from this study, could not run operationally.

**AC8:** My intention in using bias-adjusted climate model data was to demonstrate the capability of Bakaano-Hydro to operate with high-resolution climate datasets. That said, I welcome the reviewer's suggestion and will revise my experimental setup to include reanalysis data as forcing, which will also demonstrate that the framework can run with the kinds of meteorological datasets typically used in operational forecasting.

**RC9:** Operational flood forecasting models don't alert if simulations surpass thresholds computed from observations but rather if simulations surpass thresholds based on simulations, as it also has been done in Nearing et al. (2024) for GloFAS and the Google Flood Forecasting model. This removes the bias problems of simulations vs. observations and could lead to perfect flood alert rates, even if the model has a constant bias problem. Why is this even more important here? In this study, the author presents a model that performs essentially per-gauge bias correction (by inputting the routed streamflow into the TCN layer and fitting the TCN layer against observations) and then compares this to two globally calibrated operational models.

**AC9:** I acknowledge the reviewer's point that in operational forecasting systems, thresholds are typically computed from simulations rather than observations. This is done to avoid systematic biases in model magnitude and to provide internal consistency within each system. However, the purpose of this paper is not to demonstrate an operational forecasting configuration, but to provide a fair benchmarking comparison across different models. Using simulated outputs to compute thresholds independently for each model would not provide a fair basis for comparison, because models with systematic low or high bias in peak streamflow would effectively be evaluated on timing alone. For example, a model that consistently underestimates peak flows could still achieve "perfect" alert rates if its own underestimated simulations were used to define thresholds. This would artificially inflate its performance relative to a model that better matches observed flood magnitudes. By defining thresholds based on observed discharge, I ensure that both timing and magnitude are evaluated consistently across models. This approach avoids bias corrections being "baked into" the threshold definition and enables an objective assessment of which model provides more reliable flood forecasts in relation to reality. This benchmarking choice follows the reasoning that improvements in skill should be attributed to model behavior, not to calibration of evaluation metrics.

Having said that, I will take the reviewers comment into consideration in the revised version

**RC10:** Second: P12 L 12 "CSI, POD and FAR values for flood events with 1-, 2-, 5-, and 10-year return periods and 0-, 1- and 2-day timing tolerance were calculated using simulated data from Bakaano-Hydro for the period 1982 to 2016" This sentence suggests that the majority of the evaluation period for which you computed metrics are the training period of the Baakano-Hydro model (P8 L9 "The training period covered 1989–2016"). For the Google Flood Forecasting model, even for the full-run, all simulations are the result of a temporal k-fold cross validation, as stated in Nearing et al. (2024), i.e. all simulations are "unseen test data". If my understanding is correct, this renders large parts of the evaluation/conclusion unfair, as per-gauge bias corrected training data is compared to test period data from a globally calibrated model with the additional caveat from the point above.

**AC10:** I thank the reviewer for raising this point. It is correct that the metrics reported in the manuscript were computed over the full 1982–2016 period, which includes both the training (1989–2016) and validation (1982–1988) subsets for Bakaano-Hydro. This approach follows the evaluation setup used in GloFAS (Alfieri et al., 2013; Harrigan et al., 2023), where performance is reported over the full hindcast period, and was adopted here to allow direct comparability with that benchmark.

I acknowledge, however, that this creates an asymmetry with the Google Flood Forecasting model, for which all simulations are generated out-of-sample via temporal cross-validation (Nearing et al., 2024). This means the evaluation is fair with respect to GloFAS but less fair with respect to the Google AI model.

To address this, I will revise my experimental setup by introducing a cross-validation procedure for Bakaano-Hydro, ensuring that all streamflow simulations used for computing CSI, POD, and FAR are strictly out-of-sample. This will place Bakaano-Hydro on equal footing with the Google AI model while also maintaining comparability with GloFAS.

**RC11:** The metrics are not well explained or referenced. Since I wouldn't consider CSI, POD, FAR super common metrics in modeling papers of hydrology, I would recommend adding an explanation of these metrics, probably the equations, and also mention the range of these metrics and their optimal values.

**AC11:** I respectfully disagree with the reviewer's assessment that CSI, POD, and FAR are not "super common" in hydrological modeling papers. These metrics are in fact widely used in event-based flood and precipitation verification studies (e.g., Gründemann et al., 2018; Yang et al., 2021), including the evaluation of GloFAS and other global flood forecasting systems. They are standard contingency-table metrics in meteorology and hydrology for quantifying detection skill, false alarms, and overall critical success. Because these metrics are well established in the flood forecasting literature, I did not

originally include their equations. However, to avoid any ambiguity for readers who may be less familiar with event-based verification, I will add brief definitions and references in the Methods section. I do not consider it necessary to provide full derivations, since these are readily available in the cited literature, but I will ensure that interpretation and optimal ranges (e.g., POD and CSI closer to 1, FAR closer to 0) are clearly stated.

**RC12:** Figure 2: The diverging color scale, centered around 0 and with range (-1,1) does not make sense for all metrics, E.g. Alpha-NSE and Beta-KGE have their optimal value at 1. Also, please indicate what are the optimal values for each metric, which facilitates interpreting the results.
**AC12:** I agree with the reviewer and will revise the figure and the captions accordingly.

**RC13:** Figure 3/4/5: If the whiskers show the full range, why are there points outside of the whiskers? Are the whiskers maybe just indicating a certain percentile?
**AC13:** In my RainCloud plots, the whiskers do not represent the full data range. They extend only to the most extreme values within 1.5 × IQR of the lower and upper quartiles, following the standard boxplot definition. Points lying outside of the whiskers are outliers. Since all individual basin-level values are plotted in the raincloud strip, these points are explicitly shown rather than hidden, which explains why many fall outside the whiskers. I will clarify this in the figure captions.

**RC14:** Figure 6/7: They feel a bit too much and/or not very well presented, as indicating the median of model performance over all stations in a river basin only by color makes it hard for me to really get any details out of these figures. I understand the reason behind these figures (showing model differences across spatial patterns) but maybe you find a better way to visualize the results. Maybe on a map? But two pages of colored rectangles feels like an overkill in the main paper.
**AC14:** I agree with the reviewer and will revise the figure and the captions accordingly.

**RC15:** On a more general note: Figure 3/4/5/6/7 all have three columns for different flood timing tolerance. I wonder if all three of these columns for each figure are needed in the main paper or if some of the plots could be moved to the appendix. In my opinion, there are not dramatically different patterns between the columns that make it necessary to have all

three columns in the main paper.

**AC15:** I agree with the reviewer and will revise the figure and the captions accordingly.

**RC16:** P12 L9 "…using the annual maximum series (AMS) method" This sounds like it is missing a reference.

**AC16:** I will add a reference for the Annual Maximum Series (AMS) method to ensure proper attribution.

**RC17:** P12 L10f As far as I know, the paper by Nearing et al. compares GloFAS and the Google Flood forecasting model on their true forecast skill under operational settings (i.e. using only data that is available in real time) and evaluating the skill at different lead times.

**AC17:** The reviewer is correct that Nearing et al. (2024) evaluated GloFAS and the Google Flood Forecasting model under operational conditions across multiple lead times. My study is most directly comparable to their 0-day lead time (nowcast) evaluation, which is essentially a hindcast assessment based on data available up to the same day. At this lead time, both approaches assess retrospective model skill in reproducing observed flood events. I will revise the text to make clear that my evaluation relates specifically to the 0-day results in Nearing et al., and does not extend to their multi-day forecast skill analysis.

**RC18:** P22 L4 "Importantly, Bakaano-Hydro advances the state of data-driven hydrological modeling by embedding physically meaningful processes within a neural network architecture." I think I disagree with this conclusion. How was the state of data-driven hydrological modelling advanced by the findings of this paper? The presented framework does not embed physically meaningful processes in a neural network. The neural network gets the output of a physically inspired model as input. This is something fundamentally different than "embedding physically meaningful processes within a neural network."

**AC18:** I thank the reviewer for this critical point. I agree that my wording overstated the claim. The Bakaano-Hydro framework does not embed physically meaningful processes within the neural network itself. Rather, it integrates outputs from a process-based runoff model and routing scheme as structured inputs into the neural network. I will revise this sentence to clarify that the contribution is in combining process-based hydrological modeling with neural network sequence modeling in a hybrid framework, demonstrating improved reliability in data-scarce regions.